# First Investigation of Grass Carp Reovirus (GCRV) Infection in Amphioxus: Insights into Pathological Effects, Transmission, and Transcriptomic Responses

**DOI:** 10.3390/v17101367

**Published:** 2025-10-13

**Authors:** Jingyuan Lin, Meng Yang, Huijuan Yang, Guangdong Ji, Zhenhui Liu

**Affiliations:** 1College of Marine Life Sciences, Key Laboratory of Evolution & Marine Biodiversity (Ministry of Education) and Institute of Evolution & Marine Biodiversity, Ocean University of China, Qingdao 266003, China; linjingyuan2023@163.com (J.L.); yamspike@gmail.com (M.Y.); you1011you@163.com (H.Y.); jamesdong@ouc.edu.cn (G.J.); 2Laboratory for Marine Biology and Biotechnology, Qingdao Marine Science and Technology Center, Qingdao 266237, China

**Keywords:** grass carp reovirus, GCRV, amphioxus, waterborne transmission, MAPK signaling pathway

## Abstract

Amphioxus belongs to the subphylum Cephalochordata and occupies a transitional position in evolution between invertebrates and vertebrates. Due to the lack of viruses suitable for immunostimulation in amphioxus, this study for the first time explored the pathogenicity and waterborne transmission of Grass Carp Reovirus (GCRV), a double-stranded RNA virus, during its infection of amphioxus. Soaking amphioxus in GCRV suspension can cause obvious damage to gill tissues and severely disrupt the structure of gill filaments. The virus survived in seawater for no more than 48 h. Infection kinetics studies showed that the expression of VP5 (a viral capsid protein) mRNA in gill tissues peaked at 14 h. After co-culturing GCRV-infected amphioxus with healthy amphioxus for 72 h, the gills of healthy amphioxus showed obvious pathological damage. Additionally, the presence of the virus was verified by RT-PCR amplification of VP5 expression, indicating that GCRV can be transmitted via water. Transcriptome sequencing analysis showed that the Mitogen-Activated Protein Kinase (MAPK), calcium signaling pathway, and chitin metabolic pathway were significantly activated in amphioxus after GCRV stimulation. This study confirmed that GCRV can infect cephalochordates, revealing its gill-tropism and water-borne transmission ability, providing a new perspective for studying the cross-species infection mechanism of aquatic viruses and the prevention and control of aquatic diseases.

## 1. Introduction

Grass Carp Reovirus (GCRV) is a double-stranded RNA aquatic reovirus. According to the differences in its VP4 gene sequences, it can be divided into types I, II, and III. Among these genotypes, Genotype II (GCRV-II) has emerged as the predominant pathogenic genotype in current grass carp farming systems. It is widely distributed across major aquaculture regions and exhibits high virulence to both grass carp fry and adults—frequently triggering outbreaks of hemorrhagic disease, with mortality rates in fry often exceeding 80%. By contrast, Genotype I (GCRV-I), typified by the classic strain GCRV-873, was once the dominant pathogen responsible for early outbreaks of grass carp hemorrhagic disease [1]. As for Genotype III (GCRV-III), it has been rarely detected in recent epidemiological surveys and is regarded as having relatively limited pathogenicity in commercial grass carp aquaculture [2]. Currently, research on GCRV mainly focuses on fish hosts, covering areas such as genotype identification [3], functions of pathogenic proteins (e.g., VP4-mediated cellular adsorption) [4], and vaccine development [5]. GCRV can be transmitted via water bodies, and upon infection, it targets key host tissues including the gills, liver, and kidneys [6]. However, whether the host range of GCRV as an aquatic virus is limited to cyprinid fish, especially whether it can break through species barriers to infect non-fish aquatic organisms, still requires further investigation [7].

Amphioxus is a cephalochordate animal that bridges invertebrates and vertebrates. Due to its retention of primitive characteristics from vertebrate ancestors, it is regarded as a “living fossil” for studying vertebrate immune evolution [8]. Amphioxus lacks specific immune cells and antibodies, relying on innate immune mechanisms to defend against pathogens [9]. However, no viruses capable of infecting amphioxus have been identified thus far. The aim of this study are to investigate whether GCRV (used in this paper is Genotype I, with the strain designated as GCRV-Amphi2025) can infect amphioxus; and to confirm whether GCRV can be transmitted between amphioxus individuals through water.

## 2. Materials and Methods

### 2.1. Experimental Animals and Virus Strains

Amphioxus (*Branchiostoma japonicum*) was bred by our laboratory. Body mass was 1.0 ± 0.2 g. They were kept in a seawater system and water temperature of 22 ± 1 °C.

Grass Carp Reovirus (GCRV) used in this study is a double-stranded RNA (dsRNA) virus belonging to genotype I. This strain, designated as GCRV-Amphi2025 in the present study, was kindly provided by Professor Yibing Zhang’s research group at the Institute of Hydrobiology, Chinese Academy of Sciences (IHB, CAS). 

### 2.2. Infection Modeling and Sample Collection

Amphioxus were fasted for 24 h in advance. The acclimated amphioxus were divided into 2 groups (experimental group and control group), with *n* = 30 per group (3 biological replicates, *n* = 10 per replicate). Each replicate was placed in an incubator containing 200 mL of sterile seawater. For the experimental group, 40 mL of GCRV (10^5^ TCID_50_) was added; the control group received an equal volume of cell culture medium.

Samples were collected at 10 s, 2, 4, 6, 10, 14, 18, 22, 26, and 48 h after viral stimulation, respectively. At each time point, 3 biological replicate samples were collected per group (1 replicate per incubator), and each replicate included 3 amphioxus. Gill, hepatic cecum, and hindgut tissues were dissected from each amphioxus: 1 part was fixed with 4% paraformaldehyde (for histopathology), and the other was stored at −80 °C (for total RNA extraction, with 3 amphioxus’ tissues per replicate pooled as 1 RNA sample to ensure sufficient material).

### 2.3. qPCR Analysis

Total tissue RNA was extracted using the TRIzol method, and its purity was detected by NanoDrop 2000 (Thermo Fisher Scientific, Waltham, MA, USA). The RNA was reverse-transcribed into cDNA using the Total RNA Kit I (OMEGA, #R6834-01, Norcross, GA, USA). The expression of the VP5 gene was detected by the SYBR Green method (primers: VP5-F: 5′-CTCCCCGTGAGCGTGTATTT-3′, VP5-R: 5′-GTTAGCAGCGGTAGTGACTTG-3′). The reaction conditions were as follows: pre-denaturation at 95 °C for 30 s; 40 cycles of denaturation at 95 °C for 5 s and annealing-extension at 60 °C for 30 s. Each cDNA sample was analyzed with 3 biological replicates (corresponding to 3 tissue pools in Section 2.2) and 3 technical replicates per biological replicate. The relative expression levels were calculated using the ΔΔCt method.

### 2.4. Histopathology

Fixed tissues were dehydrated with gradient ethanol, embedded in paraffin, prepared as 5 μm sections, stained with HE staining and observed under the microscope. ImageJ software (version 1.54q) was used to measure the percentage of gill filament epithelial cell detachment area: 5 random fields were selected per section, and the average value of 3 sections per biological replicate was used for statistical analysis.

### 2.5. Transcriptome Sequencing

Gill tissues at 12 h post-infection and corresponding control samples (*n* = 3 biological replicates) were selected and entrusted to Novozymes for Illumina NovaSeq 6000 sequencing (PE150, single-sample data volume ≥ 6 Gb). After quality control by FastQC (version 0.11.9), raw data were aligned to the amphioxus reference genome using Hisat2 (version 2.2.1). Differential genes were screened by DESeq2 (version 1.30.1, FC ≥ 2, padj < 0.05), and GO/KEGG enrichment analysis was performed using the clusterProfiler package (version 4.16.0, Benjamini-Hochberg correction).

### 2.6. Virus Survival Assay

The virus suspension at 10^5^ TCID_50_/mL was added to sterile seawater at room temperature (25 °C). Samples were collected at 0, 2, 6, 12, 24, and 48 h. TCID_50_ was determined by EPC cytopathic effect assay, with 3 replicates set for each time point.

The Epithelioma Papulosum Cyprini (EPC) cell line was used for GCRV infectivity detection. EPC cells were cultured in Minimum Essential Medium (MEM) supplemented with 10% (*v*/*v*) heat-inactivated fetal bovine serum (FBS), 100 U/mL penicillin, and 100 μg/mL streptomycin. The cell culture environment was maintained at 28 °C with 5% CO_2_, and cells were passaged every 48–72 h when reaching 80% confluency to ensure logarithmic growth phase during the experiment.

Virus-seawater mixture preparation: The GCRV-Amphi2025 stock (10^7^ TCID_50_/mL, prepared by propagating in EPC cells as described above) was diluted with sterile filtered seawater (pre-sterilized by autoclaving) to a final concentration of 10^5^ TCID_50_/mL. The mixture was incubated at room temperature (25 ± 0.5 °C) in a sealed sterile centrifuge tube, with 3 biological replicates per time point.

Sample collection and pretreatment: At 0, 2, 6, 12, 24, and 48 h post-incubation, 1 mL of the virus-seawater mixture was collected from each replicate. Samples were centrifuged at 4 °C, 5000× *g* for 10 min to remove potential impurities, and the supernatant was collected and stored at −80 °C temporarily (tested within 24 h to avoid viral inactivation).

TCID_50_ determination via cytopathic effect (CPE) assay: EPC cells in logarithmic growth phase were seeded into 96-well cell culture plates at a density of 2 × 10^4^ cells/well, and cultured at 28 °C with 5% CO_2_ for 24 h until cells formed a uniform monolayer. The pretreated virus supernatants were serially diluted 10-fold with MEM containing 2% FBS (from 10^−1^ to 10^−8^), and the virus-free sterile seawater was used as the negative control. The original medium in the 96-well plate was discarded, and 100 μL of the diluted virus solution (or control solution) was added to each well, with 8 technical replicates per dilution. After incubating at 28 °C with 5% CO_2_ for 1 h (to allow viral adsorption), 100 μL of MEM containing 2% FBS was added to each well, and incubation was continued for 72 h. CPE was observed daily under an inverted microscope, and the wells showing typical CPE (cell rounding, detachment, and aggregation) were recorded. The TCID_50_ value was calculated using the Reed-Muench method to quantify the viral infectivity titer at each time point [10].

Data validation: For each time point, the TCID_50_ value was considered valid only if the negative control wells showed no CPE, and the coefficient of variation (CV) among the 3 biological replicates was less than 15%.

### 2.7. Waterborne Transmission Experiment

Infected amphioxus (Group A: soaked in GCRV for 14 h) and healthy amphioxus (Group B: soaked in sterile seawater) were co-cultured in sterile seawater. Each group had *n* = 15 amphioxus, divided into 3 biological replicates (*n* = 5 per replicate). Samples were collected at 12, 24, 48, 72, and 108 h post-co-culture: 3 biological replicate samples were collected per group at each time point, with 1 amphioxus per replicate. Total RNA was extracted for qPCR detection of VP5 gene expression (with 3 technical replicates per RNA sample). At 72 h, histopathological analysis was performed (3 biological replicates per group, 3 sections per replicate, consistent with 2.4).

### 2.8. Statistical Analysis

Student’s *t*-test (two-group comparison) or One-way ANOVA (multi-group comparison, Tukey’s post-hoc test) was performed using SPSS 26.0, GraphPad Prism 9 plotting, and data were presented as “Mean ± Standard Deviation (SD)” based on at least 3 biological replicates and 3 technical replicates (for qPCR, TCID_50_, and histopathology). *p* < 0.05 was considered a significant difference.

## 3. Results

### 3.1. The GCRV Strain Was Identified as Type I

GCRV can be classified into types I, II, and III, which can be identified by the triple PCR method [11]. In this study, this method was used to identify the type of GCRV (GCRV-Amphi2025) used in the experiment. The results showed that only a specific 532 bp band was amplified with the type I primer (P01-F/R) (Figure 1A), while no amplification products were obtained with the type II and type III primers. Full-length sequencing of the VP4 gene revealed that the viral strain used in this experiment had 99.78% nucleotide homology with GCRV-873 (GenBank accession number JN206664.1). In the phylogenetic tree constructed based on the VP4 sequence, GCRV-Amphi2025 clustered with GCRV-873 (bootstrap value = 98%, Figure 1B), further confirming it as a typeIGCRV.

### 3.2. GCRV Infection Caused Structural Damage of Amphioxus Gill

Currently, no virus suitable for amphioxus immunostimulation has been identified. Here, we tested two fish-targeted viruses—double-stranded RNA virus GCRV (Grass Carp Reovirus) and single-stranded RNA virus SVCV (Spring Viraemia of Carp Virus)—to stimulate amphioxus. After preliminary experiments, we found that only GCRV appeared to cause damage to amphioxus. Therefore, we conducted a detailed observation of the immune response of amphioxus to GCRV. Amphioxus were soaked in viral suspension, and gross morphological observations revealed distinct phenotypic differences: the body region (corresponding to the gill area, marked by the blue rectangle in Figure 2A) of GCRV-treated amphioxus showed obvious redness and swelling, while no such abnormality was observed in the control group (marked by the red rectangle in Figure 2A). Further histological examination (Figure 2B) demonstrated that diffuse detachment of gill filament epithelial cells, significant mesenchymal congestion, and disorganized gill tissue structure were observed in amphioxus following GCRV treatment, while gill filament epithelial cells in the control group were neatly arranged without mesenchymal congestion. Statistical analysis of damaged areas in gill tissues (Figure 2C) showed that the proportion of damaged gill tissue in the virus-stimulated group was significantly higher than that in the control group.

### 3.3. The GCRV Survived in Seawater for No More than 48 h

To determine the survival duration of GCRV in seawater, we added a 10^5^ TCID_50_/mL virus suspension to sterile seawater at room temperature (25 °C) and collected samples at 0, 2, 6, 12, 24, 48 and 72 h to detect infectivity on EPC cells. Crystal violet assay showed that the infectivity of GCRV on EPC cells after 48 h in seawater was indistinguishable from the virus-free control group (Figure 3), indicating that the virus survival time in seawater was ≤48 h.

### 3.4. Infection Kinetics of GCRV on Amphioxus

To detect the dynamic changes of GCRV in amphioxus, amphioxus were soaked in seawater containing GCRV for 2 h, then transferred to normal seawater without GCRV. During this period, gill tissues were sampled at regular intervals to detect the expression of GCRV capsid protein VP5 mRNA. Real-time fluorescent quantitative PCR results showed that the expression level of VP5 mRNA increased rapidly after amphioxus were soaked in seawater containing GCRV for 2 h, and then decreased rapidly after transfer to normal seawater without GCRV. However, it significantly increased again at 14 h, followed by a gradual decline. Although the expression was low at 48 h, it was still detectable (Figure 4A). Tissue distribution analysis showed that the expression of VP5 mRNA in gill tissues at 14 h was significantly higher than that in intestinal and hepatic caecum tissues (Figure 4B), indicating that the gills might be the main target organ for GCRV infection.

### 3.5. GCRV Can Be Transmitted Among Amphioxus Through Co-Culture

We investigated the possibility of inter-individual transmission of GCRV when infected amphioxus (Group A) were co-cultured with healthy ones (Group B). After 12 h of GCRV infection of amphioxus in Group A, they were mixed with those in Group B (Figure 5). Subsequently, gill tissues from both groups were sampled at 12, 24, 48, 72 and 108 h post-mixing to detect the expression of VP5 mRNA. After 24 h of co-culture, VP5 mRNA expression was detectable in water, but not in group B amphioxus; by 48 h of co-culture, VP5 mRNA expression in water increased, while VP5 mRNA expression in group A amphioxus decreased to a level comparable to that in water, and VP5 mRNA expression became detectable in group B amphioxus at this point. By 72 h of co-culture, VP5 mRNA expression in group B amphioxus continued to rise and was higher than that in group A amphioxus (Figure 5). By 108 h of co-culture, VP5 mRNA expression had decreased in both groups of amphioxus and in water. These data indicate that GCRV can be transmitted among amphioxus through co-culture.

Additionally, tissue sections showed obvious gill damage in both group A and group B amphioxus after 72 h of co-culture (Figure 6A); HE staining (Figure 6A) showed both groups had gill lesions, with Group A exhibiting more severe epithelial detachment and interstitial hemorrhage than Group B. Quantitative analysis (Figure 6B) revealed the damaged area was 42.6 ± 1.9% in Group A, significantly higher than 28.5 ± 0.9% in Group B (Student’s *t*–test, *** *p* < 0.001).These results confirm that GCRV can be indirectly transmitted from one amphioxus to another through water.

### 3.6. Transcriptome Analysis Reveals Significant Activation of Immune-Related Pathways

We further analyzed the effect of GCRV stimulation on gene expression in amphioxus through transcriptome sequencing. After treating amphioxus with GCRV for 14 h, gill tissues were collected for RNA-seq analysis. According to the screening criteria of |log_2_FC| ≥ 2 and padj < 0.05, a total of 523 differentially expressed genes (DEGs) were identified, of which 317 genes were significantly upregulated.

GO functional enrichment analysis showed that the DEGs were significantly enriched in biological processes such as aminoglycan metabolic process, chitin metabolic process and chitin binding, as well as molecular functions such as peptidase activity (Appendix A). KEGG pathway analysis indicated that after GCRV stimulation, the DEGs were significantly enriched in the MAPK signaling pathway, calcium signaling pathway and ECM-receptor interaction pathways (Figure 7). We validated some of the differentially expressed immunity-related genes by qPCR, and the expression of caspase-3a, caspase-3b, camk1da and camk1ga genes were all significantly upregulated, mapk9 and mapk10 genes were significantly down-regulated (Figure 8), consistent with the RNA-seq data.

## 4. Discussion

This study is the first to confirm that grass carp reovirus (GCRV)—a virus originally identified in fish—can infect the cephalochordate amphioxus. Triple PCR detection and VP4 gene sequencing results indicated that the viral strain infecting amphioxus belongs to GCRV genotype I, with its VP4 gene sharing 99.78% homology with that of the reference strain GCRV-873.

The high conservation of the VP4 protein is hypothesized to serve as a key molecular basis for GCRV’s cross-species infection capability [3]. In fish hosts, the VP4 protein of GCRV-I mediates viral adsorption by recognizing host cell surface receptors, specifically integrin αVβ3 [12] or mucopolysaccharides [13]. Our transcriptome sequencing analysis revealed significant enrichment of chitin-binding protein-encoding genes and amino sugar metabolic pathways in the gill tissues of amphioxus following GCRV stimulation. This observation suggests that GCRV may utilize chitin derivatives on the surface of amphioxus gill epithelial cells as receptors, thereby breaking through species barriers via conserved glycoprotein-mediated interactions [14]. Notably, this cross-species infection mechanism resembles that of the newly isolated HGCRV (healthy grass carp reovirus), which exhibits cross-fish-host transmission ability. This similarity highlights the need for further research to explore the universal significance of viral surface protein conservation in facilitating cross-species infection [15].

The virus survival curve in a seawater environment showed that the infectious titer dropped below the detection limit after 48 h, which differs from the survival characteristics of genotype I GCRV in freshwater environments [16]. This may be attributed to the effect of salt ions on the stability of the viral capsid [17]. Waterborne exposure experiments confirmed that the virus can be transmitted between amphioxus individuals through water [18]. Waterborne transmission may lead to the rapid spread of diseases in farmed populations [14]. However, daily water changes (≥50%) can effectively reduce the viral load in water and disrupt the transmission chain within 48 h post-infection [17].

Infection kinetics showed that VP5 mRNA in gill tissues peaked at 14 h, 3.2-fold higher than in intestinal tissues, which is consistent with the gill tropism observed in grass carp infected with genotype I and II GCRV [19]. The peak of VP5 mRNA at 14 h post-infection (hpi) results from the dynamic balance between GCRV replication and the host innate immune response: GCRV-I strains have a 12–16 h replication cycle, with capsid protein synthesis (including VP5) peaking in the late phase (8–16 hpi) to support virion assembly, which provides the viral replication basis for the 14 hpi peak [20]; meanwhile, amphioxus initiates innate immune responses (e.g., ERK2-mediated MAPK pathway activation) around 12 hpi upon viral stimulation, but the translation and functional activation of antiviral effectors require a 2–4 h delay, so the immune inhibitory effect has not yet been fully exerted at 14 hpi, allowing sustained high-level VP5 transcription—a time window shaped by both viral replication kinetics and the primitive characteristics of the amphioxus innate immune response (slightly delayed activation compared to vertebrates) [21]. HE staining of histopathological sections revealed severe damage to gill filament tissues in the infected group, accompanied by activation of the MAPK signaling pathway and the calcium signaling pathway. Since amphioxus lack T/B cells, their immune responses rely on innate pathways activated by pattern recognition receptors (such as chitin receptors) [22]. The enrichment of the ECM-receptor interaction pathway in this study may reflect attempts by gill epithelial cells to resist viral infection through extracellular matrix remodeling. However, excessive activation of the MAPK pathway may mediate apoptosis via caspase-3, leading to an imbalance between “immune defense” and “tissue damage” [6]. This response pattern is highly similar to the early inflammatory response in vertebrates [8].

This study also has limitations: first, it only uses a laboratory infection model and does not verify the viral carriage rate in wild amphioxus populations; second, it does not clarify the specific interaction mechanisms between viral and amphioxus. Future research could use yeast two-hybrid technology to screen VP4- or VP5-binding proteins, and carry out epidemiological investigations on wild amphioxus populations. In addition, single-cell transcriptome analysis can identify specific target cell types (such as ciliated cells or basal cells) in gill tissues infected by the virus, further clarifying the cell-specific mechanisms of interactions between cephalochordates and viruses.

In conclusion, this study is the first to confirm that grass carp reovirus (GCRV) can infect amphioxus and possesses the ability for waterborne transmission. These findings not only expand the known host range of GCRV but also establish a novel model for investigating the cross-species transmission mechanisms of aquatic viruses and the evolutionary trajectory of vertebrate innate immunity. 

## Figures and Tables

**Figure 1 viruses-17-01367-f001:**
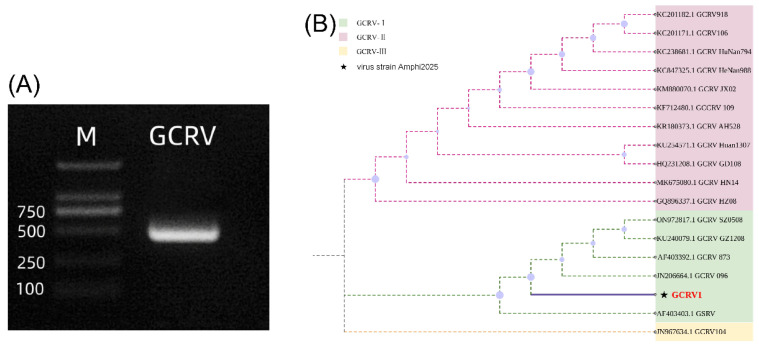
Identification of GCRV genotype I. (**A**) Triple PCR identification of GCRV genotype I. The GCRV lane shows the amplification product using genotype I—specific primers, presenting a distinct band near 500–532 bp, which confirms the virus isolate as GCRV genotype I. M represents a DNA molecular weight marker (with the positions of bands at 100, 250, 500, and 750 bp indicated). (**B**) Phylogenetic tree analysis based on VP4 gene sequences of GCRV. Different colors represent GCRV–I (light green), GCRV–II (light pink), and GCRV–III (light yellow), respectively. The virus strain Amphi2025 (marked with ★) used in this study closely clusters with GCRV–I type virus strains on the phylogenetic tree.

**Figure 2 viruses-17-01367-f002:**
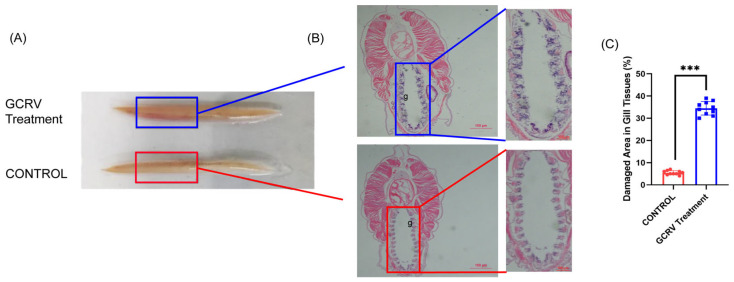
HE staining of gill tissues of amphioxus. (**A**) Gross morphology (upper panels) shows that the gill region (marked by rectangles) of GCRV-treated amphioxus exhibits obvious redness and swelling, while no such abnormality is observed in the control group. (**B**) GCRV treatment causes diffuse and detached gill filament epithelial cells, significant interstitial congestion, and disordered gill tissue structure. The letter “g” indicates gill tissue. (**C**) Statistical chart of damaged area in gill tissues (%). Each dot represents the data of a sample, demonstrating the degree of damage to gill tissues caused by infection. g: gill. *** *p* < 0.001.

**Figure 3 viruses-17-01367-f003:**
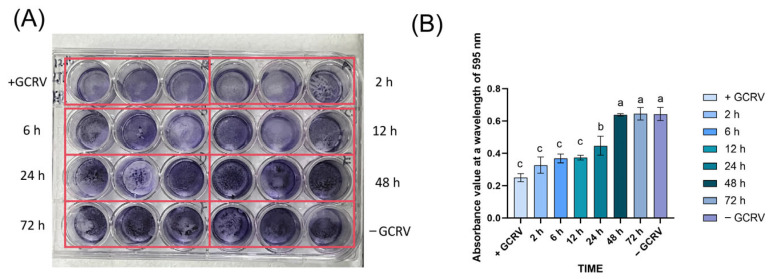
The GCRV survived in seawater for no more than 48 h. (**A**) Detection of GCRV infectious titer in sterile seawater at room temperature (25 °C) by crystal violet assay. (**B**) Detection of GCRV infectivity over time by crystal violet absorbance (595 nm). Different colors represent different time points. “+GCRV” represents the infected group, “−GCRV” is the control group, and different time points (2 h, 6 h, 12 h, 24 h, 48 h, 72 h) show the time dependent changes of the infected group. Different letters (a, b, c) are based on one-way ANOVA and Tukey’s multiple comparison test, indicating extremely significant differences between groups (*p* < 0.001).

**Figure 4 viruses-17-01367-f004:**
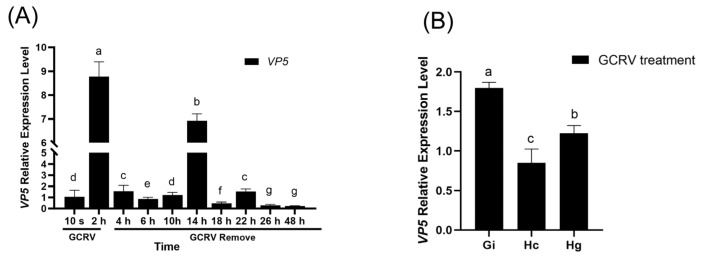
Infection kinetics of GCRV on amphioxus. (**A**) Changes in VP5 gene expression level at different time points after GCRV infection. Different letters (a, b, c, d, e, f, g) are based on one-way ANOVA and Tukey’s multiple comparison test, indicating extremely significant differences between groups (*p* < 0.001). (**B**) Relative expression level of VP5 in different tissues at 14 h post-GCRV infection. Different letters (a, b, c) are based on one-way ANOVA and Tukey’s multiple comparison test, indicating extremely significant differences between groups (*p* < 0.001). Gi: gill; Hc: hepatic cecum; Hg: hindgut.

**Figure 5 viruses-17-01367-f005:**
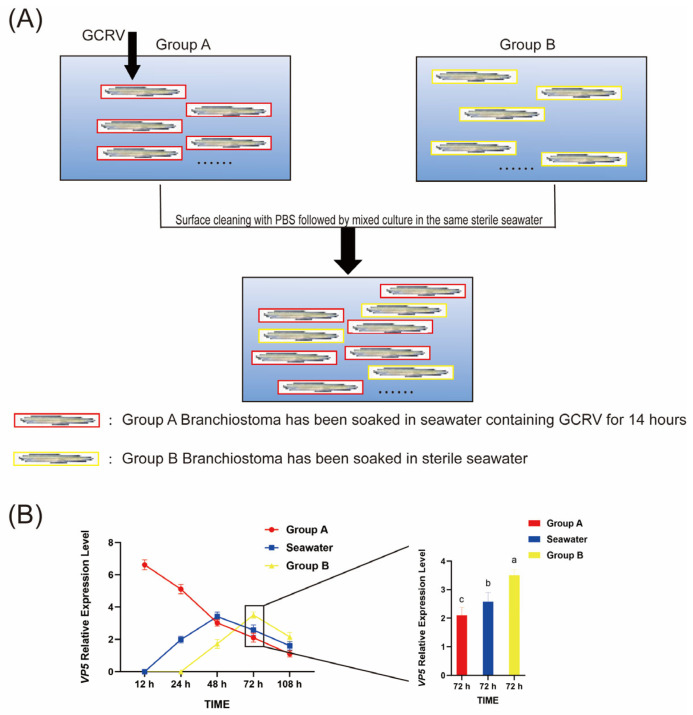
Analysis of VP5 mRNA expression in the water-borne exposure experiment. (**A**) Schematic diagram of the water-borne transmission experimental design: Group A consists of amphioxus that were pre-soaked in seawater containing GCRV for 14 h; Group B consists of healthy amphioxus that were pre-soaked in sterile seawater. After surface cleaning with PBS, the two groups were mixed and co-cultured in the same sterile seawater system to conduct subsequent sampling and detection. (**B**) Dynamic changes in VP5 mRNA expression levels: The line graph and histogram together show the relative expression levels of VP5 mRNA in Group A amphioxus, Group B amphioxus, and the co-cultured seawater at different time points (12 h, 24 h, 48 h, 72 h, 108 h) post-co-culture. At 72 h post-co-culture, the VP5 mRNA expression level in Group B was significantly 1.8 ± 0.3-fold higher than that in Group A (*p* < 0.01). Different letters (a, b, c) in the figure indicate significant differences among groups, which were determined by one-way analysis of variance (one-way ANOVA).

**Figure 6 viruses-17-01367-f006:**
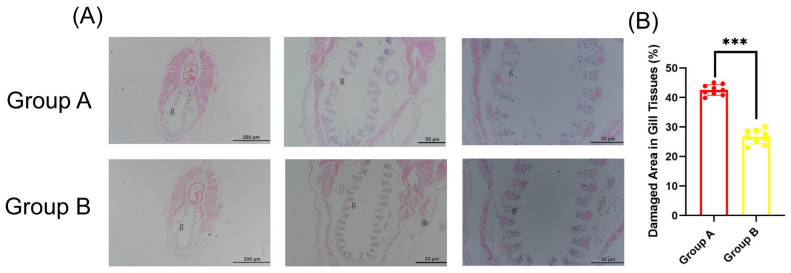
Histopathological analysis of amphioxus gill tissues. (**A**) Representative HE-stained gill sections from Group A and Group B. The letter “g” indicates gill tissue. (**B**) Quantitative analysis of damaged area in gill tissues, expressed as the percentage of damaged region relative to the total gill area. *** *p* < 0.001.

**Figure 7 viruses-17-01367-f007:**
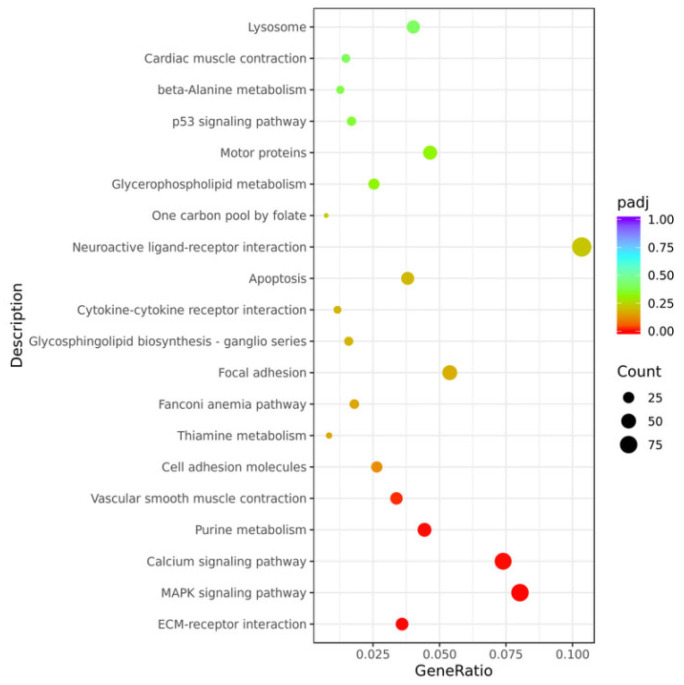
KEGG pathway analysis of gill tissues at 14 h post GCRV infection. The color of the dots represents the padj value (the redder the color, the smaller the padj value), and the size of the dots represents the number of genes (the larger the dot, the more the number).

**Figure 8 viruses-17-01367-f008:**
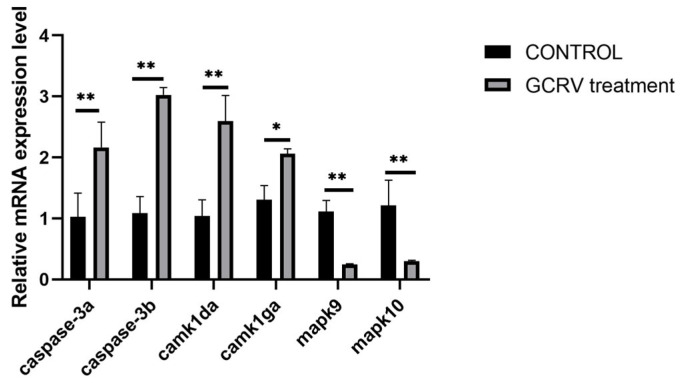
qPCR validation of differential gene expression in amphioxus following GCRV treatment. Data are presented as relative mRNA expression levels (mean ± SE) from three independent experiments. Statistical comparisons between the control group and GCRV-treated group were conducted using Student’s *t*-test. The black histogram represents the control group, and the gray histogram represents the GCRV-treated group. * indicates *p* < 0.05, ** indicates *p* < 0.01.

## Data Availability

All relevant data are available from the authors upon request and the corresponding author will be responsible for replying to the request.

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
