# Peer review of "First Investigation of Grass Carp Reovirus (GCRV) Infection in Amphioxus: Insights into Pathological Effects, Transmission, and Transcriptomic Responses"

_viruses, 2025, doi:10.3390/v17101367_

Round 1
Reviewer 1 Report
Comments and Suggestions for Authors
This study confirmed that GCRV is capable of infecting cephalochordates, thereby revealing its gill-tropism and waterborne transmission capabilities. This finding provides a novel perspective for investigating the cross-species infection mechanisms of aquatic viruses, as well as for the prevention and control of aquatic diseases. To further enhance the quality of the paper, it is suggested that the author make revisions and improvements in the following aspects.
1.The introduction of different genotypes of GCRV in the research background section is inaccurate. The prevalent pathogenic genotype of GCRV at present is GCRV II type, not GCRV I type.
2.Regarding the source of the GCRV-I strain used in this experiment, it is necessary to clarify which GCRV strain it is or to cite the relevant literature of this strain.
3.In Section 2.2 Infection Modeling and Sample Collection, it is suggested to add pictures of the external symptoms of fish infected by the virus and key data such as the cumulative mortality rate.
4.The description in the 2.6 Virus survival assay section is not specific enough. Please appropriately supplement the methods for culturing the virus on cell lines and determining the TCID₅₀.
5. In all methods sections, clearly state the number of sample replicates for each experiment (e.g., n = 3-5), such as in the histopathology section.
6.The discussion section does not explain why the peak of VP5 mRNA occurs at 14 hours. This can be discussed in combination with the virus replication cycle and the time of the host immune response to explain the expression peak.
7.Other non-standard areas: It is suggested that the case of the P value be unified throughout the text and in italics, for example, it is p in Figure 2, and P elsewhere; Check whether all the literature formats comply with the requirements of the journal. There is a repetition in reference 1.
Author Response
Reviewer 1
Comments 1:The introduction of different genotypes of GCRV in the research background section is inaccurate. The prevalent pathogenic genotype of GCRV at present is GCRV II type, not GCRV I type.
Response 1: Thanks for the comments. We have corrected the inaccurate description of GCRV genotypes in the research background section. We explicitly emphasized that GCRV-II is the currently prevalent pathogenic genotype in grass carp aquaculture, not GCRV I type.
Comments 2:Regarding the source of the GCRV-I strain used in this experiment, it is necessary to clarify which GCRV strain it is or to cite the relevant literature of this strain.
Response 2: We have clearly clarified the details of the GCRV-I strain: this strain was kindly provided by the research team of Professor Yibing Zhang from the Institute of Hydrobiology, Chinese Academy of Sciences, and was designated as GCRV-Amphi2025 in the present study. Furthermore, verification via triple PCR genotyping and VP4 gene sequencing confirmed that GCRV-Amphi2025 belongs to genotype Ⅰ of GCRV.
Comments 3:In Section 2.2 Infection Modeling and Sample Collection, it is suggested to add pictures of the external symptoms of fish infected by the virus and key data such as the cumulative mortality rate.
Response 3: We highly appreciate your valuable suggestion, which is conducive to enhancing the completeness of the experimental result presentation. We have supplemented the photos of external symptoms in virus-infected amphioxus (Figure 2). However, due to the insufficient supply of amphioxus samples in this experiment (such samples are not easy to obtain), we did not collect and analyze the mortality data. We hope to obtain this data when conditions permit in the future.
Comments 4:The description in the 2.6 Virus survival assay section is not specific enough. Please appropriately supplement the methods for culturing the virus on cell lines and determining the TCID₅₀.
Response 4: Thanks for the suggestion. We added the detail information in the revised manuscript.
Comments 5: In all methods sections, clearly state the number of sample replicates for each experiment (e.g., n = 3-5), such as in the histopathology section.
Response 5: Thanks for the correction. We added the information in the revised manuscript.
Comments 6.The discussion section does not explain why the peak of VP5 mRNA occurs at 14 hours. This can be discussed in combination with the virus replication cycle and the time of the host immune response to explain the expression peak.
Response 6: Thanks for the suggestion. We added the discussion in the revised manuscript.
Comments 7:Other non-standard areas: It is suggested that the case of the P value be unified throughout the text and in italics, for example, it is p in Figure 2, and P elsewhere; Check whether all the literature formats comply with the requirements of the journal. There is a repetition in reference 1.
Response 7: Thanks for the correction. We corrected it in the revised ms.
Reviewer 2 Report
Comments and Suggestions for Authors
Authors reported the pathogenicity and waterborne transmission of GCRV in amphioxus for the first time. The study is interesting and the manuscript designs well. However, the following issuses should be addressed before acceptance.
Minor concerns:
- GCRV can be divided into three types. Every type should be introduced, and type II GCRV is the prevalent strain.
- which type and strain of GCRV was used in the experiments? They should be in the introduction section. The donor shuld tell you the type and strain.
- Line 263, "which is consistent with the gill tropism observed in
grass carp infected with genotype I GCRV[17].", which is also consistant with genotype II GCRV. Could you mind revising it like "which is consistent with the gill tropism observed in grass carp infected with genotype I and II GCRV[17].", can supplement a literature published in virus [Type II grass carp reovirus rapidly invades grass carp (Ctenopharyngodon idella) via nostril-olfactory system -brain axis, gill, and skin on head. Viruses, 2023; 15(7):1614.].
Author Response
Comments 1:GCRV can be divided into three types. Every type should be introduced, and type II GCRV is the prevalent strain.
Response 1: We appreciate the valuable comment on clarifying GCRV genotypes. In the Introduction, we have supplemented the introduction to all three GCRV genotypes. We also explicitly emphasized that GCRV-II is the currently prevalent strain.
Comments 2:which type and strain of GCRV was used in the experiments? They should be in the introduction section. The donor shuld tell you the type and strain.
Response 2: In the Introduction, we have clearly supplemented the relevant information: used in this paper is Genotype I, with the strain designated as GCRV-Amphi2025. It was kindly provided by Professor Yibing Zhang’s research group at the Institute of Hydrobiology, Chinese Academy of Sciences. We also supplemented the identification basis of this strain to further confirm its type in this paper.
Comments 3:Line 263, "which is consistent with the gill tropism observed in grass carp infected with genotype I GCRV[17].", which is also consistant with genotype II GCRV. Could you mind revising it like "which is consistent with the gill tropism observed in grass carp infected with genotype I and II GCRV[17].", can supplement a literature published in virus [Type II grass carp reovirus rapidly invades grass carp (Ctenopharyngodon idella) via nostril-olfactory system -brain axis, gill, and skin on head. Viruses, 2023; 15(7):1614.].
Response 3: Thanks for the correction. We corrected it and added the literature in the revised ms.
Reviewer 3 Report
Comments and Suggestions for Authors
The manuscript is original, well-designed, and clearly written. The introduction provides adequate background with relevant references. The research design and methods are appropriate and sufficiently described to ensure reproducibility. Results are clearly presented, with figures and tables well-prepared, and the conclusions are fully supported by the data. The study demonstrates for the first time that GCRV can infect amphioxus, providing novel insights into cross-species viral infection and innate immune responses. The English is clear and does not require improvement. Overall, the manuscript is scientifically sound and suitable for publication with minor editorial adjustments:
-
Terminology consistency: in some sections “GCRV virus” and “GCRV” are used interchangeably (redundant), and “amphioxus” vs. “Amphioxus” capitalization is not always consistent.
-
Figures and captions: the style of figure legends is uneven, with some being very detailed and others too brief. It would be useful to harmonize them and ensure that all symbols (e.g., ★, letters a–g) are clearly explained.
-
Reference formatting: some entries in the References contain repeated titles or minor errors (e.g., “Functional Characterization of Interleukin 17 Family Members and Their Receptors in amphioxusFunctional Characterization…”).
-
Language polishing: the English is good, but certain sentences could be streamlined for readability (e.g., “the GCRV virus appeared to cause damage” → “GCRV caused damage”).
-
Typographical details: spacing and symbols should be standardized (e.g., “25 ℃” should be “25 °C”; units like µm, ± should be consistently formatted).
.
Author Response
Comments 1:Terminology consistency: in some sections “GCRV virus” and “GCRV” are used interchangeably (redundant), and “amphioxus” vs. “Amphioxus” capitalization is not always consistent.
Response 1: Thank you for your correction. We have corrected the above in the revised manuscript.
Comments 2:Figures and captions: the style of figure legends is uneven, with some being very detailed and others too brief. It would be useful to harmonize them and ensure that all symbols (e.g., ★, letters a–g) are clearly explained.
Response 2: Thank you for your suggestion. We have revised and improved the figure legends in accordance with your comments. Specifically, in Figure 4, we have explicitly listed the meanings of items such as A, B, C, etc.
Comments 3:Reference formatting: some entries in the References contain repeated titles or minor errors (e.g., “Functional Characterization of Interleukin 17 Family Members and Their Receptors in amphioxusFunctional Characterization…”).
Response 3: Thank you for your correction. We have corrected the above in the revised manuscript.
Comments 4:Language polishing: the English is good, but certain sentences could be streamlined for readability (e.g., “the GCRV virus appeared to cause damage” → “GCRV caused damage”).
Response 4: Thank you for your comments. We have carefully proofread and polished the language throughout the manuscript; additionally, we have invited Dr. Yujun Liang, who previously worked at Yale University, to revise the manuscript’s language.
Comments 5:Typographical details: spacing and symbols should be standardized (e.g., “25 ℃” should be “25 °C”; units like µm, ± should be consistently formatted).
Response 5: Thank you for your correction. We have corrected the above content in the revised manuscript.